# Emerging Arthropod-Borne Viruses Hijack the Host Cell Cytoskeleton During Neuroinvasion

**DOI:** 10.3390/v17070908

**Published:** 2025-06-26

**Authors:** Flora De Conto

**Affiliations:** Department of Medicine and Surgery, University of Parma, 43126 Parma, Italy; flora.deconto@unipr.it

**Keywords:** cytoskeleton, antiviral therapy, encephalitis, neurological disorders, arthropod-borne viruses, emerging zoonoses

## Abstract

Arthropod-borne viral infections, ranging from asymptomatic to fatal diseases, are expanding from endemic to nonendemic areas. Climate change, deforestation, and globalization favor their spread. Although arboviral manifestations mainly determine the onset of generalized symptoms, distinct clinical signs have been assessed, depending on the particular arthropod-borne virus (arbovirus) involved in the infectious process. A number of arboviruses cause neuroinvasive diseases in vertebrate hosts, with acute to chronic outcomes. Long-term neurological sequelae can include cognitive dysfunction and Parkinsonism. To increase knowledge of host interactions with arboviruses, in-depth investigations are needed to highlight how arboviruses exploit a host cell for efficient infection and clarify the molecular alterations underlying human brain diseases. This review focuses on the involvement of host cytoskeletal networks and associated signalling pathways in modulating the neurotropism of emerging arboviruses. A better understanding at the molecular level of the potential for emerging infectious diseases is fundamental for prevention and outbreak control.

## 1. Introduction

The growth of human populations, attendant deforestation, intensive agriculture, and climate change have been the leading causes of the emergence of infectious diseases in the last few years. Most emerging pathogens causing human infection are zoonotic [1]. Most zoonotic diseases are highly virulent, causing billions of cases and millions of deaths every year worldwide [2,3], and as a result, they represent public health threats and economic burdens [4,5,6,7,8].

Arthropod-borne viruses (arboviruses) are RNA viruses transmitted through the bites of infected arthropods (mosquitoes and ticks) during blood feeding. This group of viruses comprises at least 150 members related to human disease and poses a substantial health threat, especially in tropical and equatorial countries [9,10].

In recent years, arthropod-borne virus infections have significantly increased in both endemic and new areas [11,12,13,14]. The arrival of arboviruses in new areas facilitates the emergence of epidemic outbreaks due to the high susceptibility of certain individuals. In general, the expansion of arbovirus outbreaks depends on several factors, including climate change (increased temperatures and high variability in precipitation patterns), globalization, trade, socioeconomics, viral genomic evolution, and potential virus adaptation to new vectors [15,16,17,18,19]. Most of the arboviral epidemics of the past 75 years have been from members of the *Flaviviridae* and *Togaviridae* families [20].

Although the published literature provides extensive data on arbovirus characteristics, an accurate, in-depth review of the host cell factors co-opted during the infectious process and the mechanisms by which they cause neural dysfunction is lacking.

This review focuses on the involvement of the cellular cytoskeleton during the arbovirus infection of neuronal cells. The issues presented could constitute a valuable premise for future in-depth studies and the possible discovery of innovative therapeutic approaches.

## 2. Emerging Neurotropic Arboviruses

Most zoonotic virus infections are severe and neuroinvasive in non-natural hosts [21]. It must be considered that zoonotic transmission can favor the emergence of new arbovirus strains with neuroinvasive potential. In this regard, it has been assessed that several emerging (and re-emerging) arboviruses can infect the brain, meninges, and spinal cord, causing relevant neurological signs, such as confusion, seizure, memory impairment, and long-term defects in the central nervous system (CNS) [20,22,23].

The following are the principal characteristics of the most relevant emerging encephalitic arboviruses which cause neuroinvasive human diseases (see Table 1).

In 2018, the RVV and ZIKV viruses were identified as two of the eight pathogens of greatest concern, given the likelihood of them causing a future epidemic. To date, there are insufficient countermeasures for these viruses, which only underlines the urgent need for further research and investment [17].

There are currently no approved antivirals for neurologic infections with arboviruses. Considering that treatments are principally supportive, specific antivirals, treatments for post-infection sequelae, and preventive vaccines to protect the CNS from damage caused by viral infection need to be developed.

## 3. Neuroinvasion and the Neuropathogenesis of Arboviruses

Most arboviral infections initiate at epithelial or endothelial surfaces during blood feeding by mosquitoes and ticks. Having evaded the immune system’s defences, arboviruses may spread to other tissues with increased replication and the over-activation of their host’s innate immune responses. Shortly after viraemia, neuroinvasive arboviruses may hijack the CNS by means of different mechanisms, such as infection of olfactory sensory neurons in the olfactory neuroepithelium, retrograde transport of the virus along axon microtubules, exploiting infected leukocytes as Trojan Horses, and invasion of the blood–brain barrier. Such events cause severe effects on the brain, including meningitis, encephalitis, and death.

It is known that after arbovirus infection, a complex pattern of recognition receptors activates the innate immune response. Among these, it has been postulated that the Toll-like receptors TLR3 and TLR7 may have either pathogenic [44] or defensive rules [45] during WNV infection in the CNS. Moreover, ZIKV activates TLR3 in neural progenitor cells, promoting apoptosis mechanisms and dysregulating cell physiology [46]. In this scenario, it has been assessed that retinoic acid-inducible gene-I-like receptor signalling restricts WNV replication in the CNS [47]. Considering that neurons are irreplaceable, the activation of immune responses counteracts the onset of extensive inflammation and cytotoxic effects.

The blood–brain barrier, composed of astrocytes, pericytes, and brain microvascular endothelial cells, plays a key role in the pathogenesis of neurotropic viruses by mediating their access to the CNS [21]. Neurotropic viruses have evolved very sophisticated mechanisms for evading the defences of the blood–brain barrier. Among these are changes in the expression and phosphorylation levels of the proteins constituting the tight junctions, alterations to actin filaments, and disruption of the basal lamina [21].

Neurons are the main target of encephalitic arboviruses, although other cell types of the CNS can also be infected [9]. This is the case for TBEV, which can infect the astrocytes located between synapses and endothelial cells, causing severe sequelae for the functioning of the CSN. Thought-provoking experimental data have shown that TBEV can infect rat astrocytes by subverting the actin cytoskeleton, without, however, affecting their vitality [48]. Then again, these data are in contrast with those observed in human glioblastoma cells, where microtubules undergo profound rearrangements following TBEV infection [49]. These observations have revealed that the interaction of the virus with each host cell is strictly peculiar, and, given the greater resilience of astrocytes to viral infection, these cells have been identified as possible reservoirs for the spread of infection.

Among the viral determinants of neurovirulence are arboviral envelope proteins and the proteins codified by the 3′ untranslated region and regulatory non-structural proteins [50,51,52]. Neuronal cell damage may occur either by the direct effects of viral replication or uncontrolled immune responses. Specifically, arboviruses are able to activate cellular factors which cause the death of neuronal cells and, at the same time, stimulate the activation of the inflammatory responses responsible for further exacerbation of the damage. Infected neurons exhibit typical features of apoptosis, with consequential neuronal degeneration and necrosis in the CNS [53,54]. It has been mooted that the Zika virus NS2B3 protease cleaves septin-2, a cytoskeletal factor involved in cytokinesis [55]. The expression of NS2B3 protease mediates neuronal toxicity, resulting in delayed cell division, increased apoptosis, and multipolar spindles. In addition, the neuropathogenesis of arboviruses has also been associated with neuroinflammation due to the release of chemokines and cytokines.

Although these data attest to the neuroinvasive ability of arboviruses, which leads to impairment of the CNS, various aspects of arbovirus pathogenesis in the human brain remain poorly understood.

## 4. Neurotropic Arboviruses Hijack the Host Cell Cytoskeleton for Efficient Replication

Little is known about the molecular mechanisms underlying the cell biology of arbovirus infection since experimental infections carried out in cell models are limited. Understanding the relationship of neuroinvasive arboviruses with the host cytoskeleton could shed light on key aspects connected to their pathogenic action that have not yet been clarified.

The cytoskeleton of eukaryotic cells is a highly dynamic filamentous network which radiates throughout the cell, including the following three interconnected filament types: actin filaments (microfilaments) (AFs), microtubules (MTs), and intermediate filaments (IFs) [56]. The cytoskeleton is involved in numerous cellular functions, such as the intracellular movement of different molecular cargos, cell morphology regulation, cell migration, apoptosis, cell differentiation, and cell division [57,58,59,60,61].

Viruses hijack the host cell machinery to ensure their entry, replication, transport, and viral progeny release. To this end, viruses have evolved sophisticated mechanisms for modulating the host cytoskeleton characteristics to their advantage [56]. However, it has yet to be solidly demonstrated that the role played by the cytoskeleton during viral infection is clear, considering that it could also contribute to regulating the host’s innate immune response against viral infection [62].

Previous data have proposed that viruses primarily use AF and MT networks to accomplish their life cycles [63,64]. Moreover, the motor proteins dynein and kinesin promote viral transport along MTs and myosin transport along AFs.

As for the role of the cytoskeleton during neurotropic arbovirus infection, it has been suggested that the disruption of AFs facilitates ZIKV and WNV infection in the CNS [21]. A significant reorganization of the actin network has been found during WNV infection through intervention of the NS1 protein [65]. Tunnelling nanotubes, which are actin-rich projections forming intercellular bridges, have been observed, facilitating the transfer of WNV virions. The observed actin modulation was cell-type-dependent, having been seen in epithelial cells but not in neuronal and glial cells. In addition, it has been shown that AFs exert a critical role in WNV egress from the host cell [66].

Additionally, ZIKV induces actomyosin cytoskeletal rearrangements as a prerequisite for increasing cell contractility and migration, which favors the progression of virus replication [67]. Moreover, MTs play a relevant role during ZIKV infection. Accordingly, mispositioned spindles and augmented centrosome numbers may contribute to microcephaly caused by ZIKV infection [68,69]. It has been shown that ZIKV drastically reorganizes MTs and cytokeratin 8, forming new molecular structures surrounding the viral replication complexes [70]. ZIKV also disrupts the centrosome architecture and recruits the ubiquitin ligase DTX4 to further the degradation of TBK1, a regulator of the innate immune response [71,72,73]. As a result, ZIKV-infected cells express low levels of interferon beta. Moreover, the ZIKA NS5 protein reorganizes MTs, stimulating their acetylation to overcome the innate defence mechanisms controlled by tubulin-deacetylase HDAC6 [74].

Both GTPase dynamin and MTs mediate the internalization of WNV through the endocytosis of tight junction membrane proteins, such as claudin-1 and JAM-1, in epithelial and endothelial cells. This event has not been observed in DENV infection, suggesting different mechanisms for WNV neuroinvasion [75]. Likewise, it has been observed that MT disruption through nocodazole treatment negatively interferes with WNV axonal retrograde transport [76].

DENV interacts with AFs in the early phases of its replication [77]. More specifically, Rac1 signalling activation regulates AFs, modulating DENV cell entry and exit [72]. Additionally, DENV activates the P13K/Akt/Rho GTPases signalling pathway at the early stages of infection, causing the reorganization of actin and promoting viral replication [78]. In particular, DENV infection can increase microglial cell migration, and clathrin- and/or actin-disruption has retarded this effect [79]. Concerning MTs, DENV recruits them and the associated motor proteins for its cytoplasmic transport. The replication activity of DENV has also been linked to a profound vimentin reorganization and phosphorylation [80,81]. Moreover, the centrosome and the cytoskeleton were identified among the interacting protein partners of DENV [82].

JEV enters neuronal cells by an endocytosis mechanism, requiring a huge actin network reorganization activated by RhoA and Rac1 signalling [83,84]. Also, PIXV is facilitated in cell entry by the chemically induced disorganization of AFs [85]. On the other hand, the inhibition of MT polymerization and MT stabilization do not interfere with PIXV replication.

CHIKV infection in human muscle cells has shown that the expression of several cytoskeletal proteins, such as desmin, actin, vimentin, stathmin 1, and lamins, is differentially regulated during the infectious process, highlighting their involvement in the viral life cycle [86]. Moreover, CHIKV induces the rearrangement and reorganization of vimentin around viral replication complexes.

Both the early and late stages of RVV infection occur at the apical and basolateral membranes of polarized epithelial cells, where they cause the disorganization of AF and MT networks, facilitating virion release [87]. RVV targets the actin cytoskeleton at the transcriptional and cellular levels, causing occurrence of the cytopathic effect, which contributes to its pathogenicity [88].

In general, the intervention of the cytoskeleton is indispensable for flavivirus replication [70,77]. Following this, it was recently demonstrated that the proviral SAD1/UNC84 domain protein 2 (SUN2), by linking the cytoskeleton and nucleoskeleton, facilitates their reorganization, promoting viral RNA synthesis [89]. In addition, nesprin proteins, in connecting SUN2 to the cytoskeleton, are involved.

Figure 1 summarizes the main cytoskeletal changes observed during arbovirus infection.

Confocal microscopy studies focusing on the ability of arboviruses to reorganize the cytoskeleton have evidenced its involvement in the infectious process. In this scenario, the interaction between the non-structural protein 4A of DENV and cellular vimentin induced both relevant intermediate filament rearrangements and vimentin phosphorylation [81]. In addition, significant alterations in actin assembly and expression were observed during the DENV infection of human endothelial cells [90].

The reports mentioned above suggest that arboviruses can subvert complex host cytoskeletal organization to create an environment which favors their replication and spread. Since arboviruses can target different host cell types, leading to diverse consequences, it would seem relevant to analyze the co-opted host cytoskeletal functions of specific cell types in the future.

## 5. Modulation of Cytoskeletal Protein Expression During Arbovirus Neuroinvasion

Virus–host interactions involve complex molecular and signalling pathways that are mainly dependent on the lineage of the infected host cell. To decipher the cell biology mechanisms underlying arbovirus infection in the CNS, a kinetic analysis of the host’s proteome modulation can reveal significant differences in protein levels, turnover rates, and expression patterns. Moreover, studies identifying virus–host protein interactions can provide insights into the use of host proteins to assist the virus in replication and immune evasion. In this respect, changes in cytoskeleton dynamics have been assessed during CHIKV infection before and after the appearance of neurological clinical symptoms [91]. Specifically, CHIKV infection induces an early shut-off of host cytoskeletal protein expression, followed by its up-regulation during the appearance of clinical neurological signs. Observations based on the proteome alterations in a mouse model have suggested that most of the aspects of CHIKV infection, such as disease severity, neurological complications, disease susceptibility, and immunocompetence, can be better dissected [92].

The up-regulation of specific genes (ERV316A3_12q24.13, ERV316A3_3q27.3e, ERV316A3_7q34a, ERVLB4_12p13.2a, HARLEQUIN_17q12, HERV4_4q22.1, HERV9_11q21, HERVK11D_2q11.2, HML3_12q13.12, HML3_16p13.3, HML6_14q24.2, HML6_19p13.2c, HUERSP3_2p25.2, MER101_1p22.2a, and MER101_2p25.2) belonging to human retrotransposable elements of retroviral origin (HERVs) has been assessed through analyses of human astrocytes infected by CHIKV, MAYV, and OROV [93]. Specifically, the genes concerned regulate different cellular processes, such as those of the cytoskeleton. HERV up-regulation, for example, could modulate the inflammatory response, promoting viral replication and the occurrence of neurological disorders.

Profound modifications to the host proteome following different time-points of WNV infection have been assessed [94]. Specifically, the up-regulation of restin, an IF-associated protein, contributes to neuronal apoptosis.

Quantitative proteomic approaches to mouse brain tissue have shown modifications to cytoskeletal protein expression in the early stages of WNV replication, which contribute to cytoskeleton maintenance [95]. In this regard, several proteins related to the AF cytoskeleton and the Rho GTPase signalling pathway are significantly modified during WNV neuroinvasive infection.

A quantitative analysis of protein phosphorylation during virus infection has provided crucial information about the molecular mechanisms of viral pathogenesis. This analysis showed a critical role for c-Jun N-terminal kinase in neuroinflammation induced by JEV [96]. Among the phosphorylated cytoskeletal proteins observed is the one chiefly involved, actin, showing its importance during the infectious process.

Table 2 shows the most relevant modifications to the host’s cytoskeletal proteome observed during the arbovirus infection of neuronal cells in rodent models. These issues represent an important premise for further evaluation of early biomarkers for the prevention of severe neurological diseases as well as for diagnostic purposes.

## 6. Discussion

The ongoing emergence of new zoonotic diseases presents a significant challenge for public health, urging the swift development of an effective surveillance system for their prevention and management.

Deforestation, intensive agriculture, and climate change have been the primary factors contributing to the spread of arboviruses from endemic to nonendemic areas in recent years. Notably, in 2022, the World Health Organization defined a prevention and preparation plan for arboviral pandemics [97].

Arboviruses have an increased likelihood of faster adaptation to environmental and other changes, such as shifts in vectors and hosts, which can lead to significant health and socioeconomic issues. Moreover, this is taking place under the current dearth of specific antiviral treatments for a great majority of arbovirus infections [98].

Since arboviruses cause recurrent epidemics worldwide, primarily resulting in severe neurologic complications, understanding the mechanisms that lead to neural dysfunction serves as a valuable foundation for developing innovative broad-range antiviral strategies. To enhance our capacity to proactively prevent and manage emerging arbovirus infections, it is vital to explore the complex biological mechanisms regulating them and the potential interconnections between humans, animals, and the environment from a One Health perspective. A thorough understanding of arbovirus pathogenesis is essential for developing early biomarker and new therapeutic targets aimed at preventing severe neurological diseases and facilitating rapid diagnoses. Molecular-level analyses designed to dissect protein interactions could provide a solid basis for examining the molecular mechanisms of neuroinvasive arboviral diseases. Specifically, it is crucial to identify the cellular factors that arboviruses exploit to their advantage for genome replication, viral protein production, and virion assembly. In this context, recent data have highlighted efforts to establish host-directed therapeutics against arboviruses. Specifically, an inhibitor of Sec61, an essential host factor for viral proteostasis, has been used in cellular models of SARS-CoV-2, influenza A virus, and flavivirus (Zika, West Nile, and Dengue virus) infections, impacting viral replication and demonstrating potential broad-spectrum activity [99]. Furthermore, a pronounced and sustained antiviral activity against various RNA-enveloped viruses has been evaluated for the natural compound thapsigargin, offering a promising therapy with host-directed antiviral properties [100].

The cell cytoskeleton forms a barrier for viral infection, impacting virus invasiveness and pathogenesis. Numerous cytoskeletal elements can modulate the end of the virus–host interaction. Among these, the lamina of the nuclear cytoskeleton regulates viral trafficking to the nucleus. Consequently, viruses have developed sophisticated exploitation mechanisms to overcome this barrier, such as activating viral and cellular kinases to promote lamina phosphorylation and subsequent disaggregation, facilitating their nuclear ingress and egress [101]. Additionally, the vimentin network is another target for viral replication since this cytoskeletal component interacts with several cellular elements, regulating diverse signalling mechanisms. In particular, the presence of vimentin outside of a cell contributes to the pathogenesis of infectious diseases by modulating the initial steps of viral infection. Overall, research has demonstrated that vimentin regulates various stages (virus adsorption, entry, replication, and release) of the viral life cycle, with the capacity to swiftly adjust its expression levels [102]. Furthermore, the complex network of actin filaments, which provides support to the subapical membrane and stabilizes intercellular junctions in polarized epithelial cell monolayers, poses an obstacle for many viruses during host invasion [103]. In response, viruses have evolved sophisticated strategies to dismantle the cortical actin meshwork, thereby aiding the infectious process.

In general, while the specific mechanisms of cell invasion used by different viruses can vary, a highly dynamic state of the cytoskeleton is necessary to support the infectious process [62]. In this regard, it has been evidenced that diverse neurotropic arboviruses modulate the expression of different cytoskeletal proteins, as shown in Table 2. Moreover, it has been shown that neurotropic flaviviruses, such as WNV, co-opt specific cytoskeletal components in a different way than haemorrhagic flaviviruses, suggesting the activation of selective molecular mechanisms to ensure efficient neuroinvasion. Specifically, a severe WNV infection involves the crossing of polarized cell layers to gain access to organs and the CNS. To this end, WNV, but not the serotype 2 of DENV, affects the barrier function of tight junctions, boosting the endocytosis of some membrane proteins, such as claudin-1 and JAM-1, in epithelial and endothelial cells [75]. Importantly, this latter event is regulated by the cytoskeletal GTPase dynamin and microtubule-based transport.

In the case of JEV infection, it has been shown that entry is independent of clathrin, while the regulatory proteins of actin filaments, such as RHOA, RAC1, proteins of the ARP2/3 complex, and the N-WASP family (LIMK1, PAK1, and ROCK2), are crucial for the establishment of infection [104]. In addition, in mouse models, some CNS genes are differentially regulated during JEV infection. Interestingly, 80% of the inducible genes are also activated by the Sindbis virus, suggesting, in this case, the existence of common signalling mechanisms despite diverse virus life cycles [105]. Notably, one of the inducible genes codes for a microtubule-associated protein, envisaging its regulatory relationship with microtubule dynamics.

The reports analyzed in this review attest to the significant implication of cytoskeletal components in regulating arbovirus infection in the CNS. In particular, AFs and the associated regulatory factors appear crucial in modulating the very early phases of viral infection. These data are also supported by recent observations of the proteomic profile in the sera of subjects with fatal CHIKV infection; in this case, an up-regulation of actin and associated proteins (vinculin, profilin 1, actin gamma 1, and actinin alpha 1 and 2) has been observed [106]. Moreover, it has been shown that MTs and related kinetoproteins are recruited by arboviruses for efficient cytoplasmic transport. In addition, MTs and IFs undergo a profound reorganization for efficient virus replication and the modulation of innate defence mechanisms. Notably, during viral replication, several cytoskeletal components and related signalling pathways became up-regulated, as shown in Table 2. These observations assess the ability of neurotropic arboviruses to co-opt the host cytoskeleton of different cell lineages for efficient replication and neuroinvasion.

Although it is clear that most of the mechanisms adopted by neurotropic viruses are virus-dependent and involve specific cytoskeletal proteins, there are limited data on the viral proteins that exert the observed regulatory function. Therefore, further investigations will be needed in the future.

One limitation of this review is the paucity of published data on the human brain in the literature, conceivably due to the difficulty of undertaking in vivo and in vitro experiments and also given the high level of biosafety required. Accordingly, as reported in this review, rodent models have been largely employed to assess the molecular mechanisms associated with infections by neurotropic arboviruses. However, any conclusions drawn must be cautious, considering that human and rodent CNSs show remarkable differences when it comes to their cellular and molecular compositions, gene expression, and interactions between different cells [107]. On this basis, recent reports have suggested the employment of human brain organoids as an alternative research model [108]. Moreover, human brain organotypic cell cultures may represent a useful ex vivo model derived from human brain tissue, either obtained post-mortem or by surgical resection. However, for this latter model, it needs to be considered that there is limited access to human brain tissue and there is a need for oxygen availability for cell maintenance in cultures due to the absence of blood flow. These aspects point to the fact that, although promising, these models still need to be refined.

In conclusion, additional insights into viral–host interactions in cell culture systems and ex vivo models are required to expand our understanding of the cell biology of arbovirus-infected cells and better dissect the neuropathogenesis mechanisms in the human CNS at the molecular level. In this regard, the recent and innovative multi-omics approaches (genomics, transcriptomics, metabolomics, and proteomics), focusing on viruses, virus–host interactions, and disease dynamics, will represent a relevant research approach for further in-depth analyses.

## Figures and Tables

**Figure 1 viruses-17-00908-f001:**
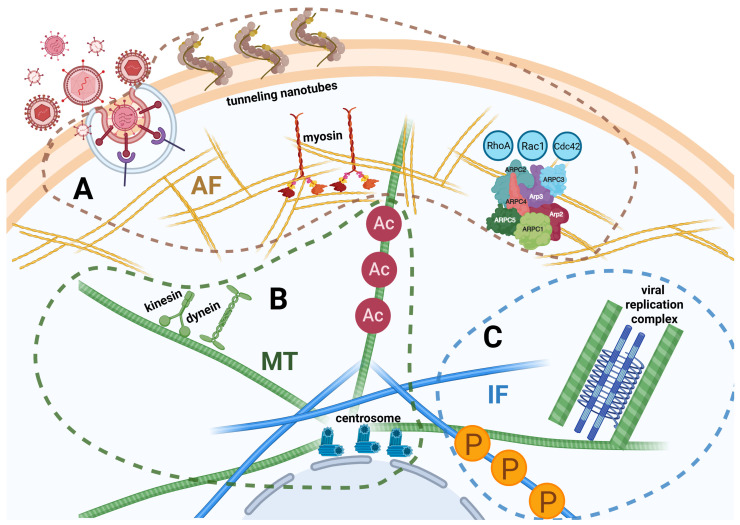
Representation of the main elements of arbovirus-induced cytoskeleton remodelling. (**A**) A virus binding to the cell receptor activates the GTPases RhoA, Rac1, and Cdc42, stimulating actin filament (AF) and remodelling for efficient virus entry. Kinetoproteins (myosin) are recruited. The viral infection stimulates the development of tunnelling nanotubes, which are actin-rich projections forming intercellular bridges. (**B**) Enhanced posttranslational acetylation regulates MT dynamics, cytoplasmic transport, and innate defence mechanisms. Motor proteins (kinesin and dynamin) are recruited. Mispositioned spindles and augmented centrosome numbers contribute to neuropathogenesis. (**C**) Changes in vimentin expression and posttranslational phosphorylation contribute to viral replication and signalling. MTs and cytokeratin undergo relevant reorganization to form structures surrounding the viral replication complexes. IF: intermediate filaments. The three viral impact mechanisms have been denoted by the dashed lines of different colors (A: brown; B: green; C: sky blue). Created in Bio Render. De Conto, F. (2025) https://BioRender.com/egjxqmw. Accessed on: 29 May 2025.

**Table 1 viruses-17-00908-t001:** Most relevant features of emerging encephalitic arboviruses causing neuroinvasive human diseases.

Virus	Family	Vector	Other Routes of Transmission	Diseases	References
Dengue virus (DENV)	*Flaviviridae*	Mosquito		Dengue fever, severe shock syndrome, and endothelial disfunction	[24,25,26]
Zika virus (ZIKV)	*Flaviviridae*	Mosquito (mainly *Aedes aegypti*)	Mother to foetus, sexual contact, blood transfusion, and organ transplantation	Microcephaly in neonates, Guillain-Barré syndrome, radiculomyelitis, and meningoencephalitis	[27,28]
West Nile virus (WNV)	*Flaviviridae*	Mosquito (*Culex* spp.)		Meningitis, encephalitis, and paralysis cognitive dysfunction	[29]
Japanese encephalitis virus (JEV)	*Flaviviridae*	Mosquito (*Culex* spp.)	Oral shedding	Acute encephalitis	[30,31]
Saint Louis encephalitis virus (SLEV)	*Flaviviridae*	Mosquito (*Culex* spp.)		Neuroinvasion is more frequent in immunocompromised and solid-organ-transplanted subjects	[32]
Tick-borne encephalitis virus (TBEV)	*Flaviviridae*	*Ixodes* spp.		Meningitis and encephalitis, cognitive dysfunction, and memory impairment	[33]
Powassan virus (POWV)	*Flaviviridae*	*Ixodes* spp.		Encephalitis and meningitis	[34]
Chikungunya virus (CHIKV)	*Togaviridae*	Mosquito (most frequently *Aedes aegypti* and *Aedes albopictus*)		Neurological manifestations and haemorrhagic diseases	[35,36,37]
Eastern equine encephalitis virus (EEEV)	*Togaviridae*	Mosquito		Encephalitis	[38]
Mayaro virus (MAYV)	*Togaviridae*	Mosquito *(Haemagogus* spp.)		Myalgia, rash, and neurological issues	[39]
Pixuna virus (PIXV)	*Togaviridae*	Mosquito		Severe encephalomyelitis	[40]
Rift Valley virus (RVV)	*Phenuiviridae*	Mosquito		Severe neurological manifestations (meningoencephalitis)	[41]
Oropouche virus (OROV)	*Peribunyaviridae*	Biting midges and some mosquitoes		Encephalitis and meningoencephalitis fever, headache, myalgia, and arthralgia	[42,43]

**Table 2 viruses-17-00908-t002:** Main expression modulations of the cytoskeletal proteins and associated signalling pathways during arbovirus infection of the CSN.

Cytoskeletal Protein	Cytoskeleton-Associated Signalling Pathways/Cytoskeleton Regulatory Proteins	Host	Virus	Role	Effects	References
	Restin	Rat (cortical neurons)	WNV	Inhibits cell proliferation to induce apoptosis by binding to tropomyosin	Up-regulation	[94]
Actin and Tubulin	Dynamin-1	Mouse (brain)	WNV	Cytoskeleton organization	Up-regulation	[95]
MAP1B and MAP2		Mouse (brain)	WNV	Nervous system development and neurogenesis	Down-regulation	[95]
	Rho GTPase signalling	Mouse (brain)	WNV	Actin remodelling and clathrin-mediated endocytosis activation	Up-regulation	[95]
	Dynamin-1	Mouse (brain)	CHIKV	Clathrin-mediated endocytosis activation	Up-regulation	[91]
	Rho GTPase signalling	Mouse (brain)	CHIKV	Actin remodelling	Up-regulation	[91]
	ARPCB1	Mouse (brain)	CHIKV	MT filament formation	Up-regulation	[91]
TUBB3		Mouse (brain)	CHIKV	Cytoskeleton organization	Up-regulation	[91]
	CORO2B and MYPT1	Mouse (brain)	CHIKV	Cytoskeleton rearrangement/motility	Up-regulation	[91]
	CORO1A	Mouse (brain)	CHIKV	Functioning in the invagination of plasma membrane and in forming the protrusions of the plasma membrane involved in cell locomotion	Up-regulation	[91]
Tubulin		Mouse (brain)	CHIKV	Cytoskeleton organization	Up-regulation	[91]

Legend: ARPCB1, actin-related protein 2/3 complex subunit 1B; CORO1A, coronin 1A; CORO2B, coronin 2b actin-binding protein; MAP1B, microtubule-associated protein 1B; MAP2, microtubule-associated protein 2; MYPT1, protein phosphatase 1 regulatory subunit mediating binding to myosin; TUBB3, tubulin beta-III.

## Data Availability

No datasets were generated or analyzed in the current study.

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
