# Peer review of "Emerging Arthropod-Borne Viruses Hijack the Host Cell Cytoskeleton During Neuroinvasion"

_viruses, 2025, doi:10.3390/v17070908_

Round 1

Reviewer 1 Report

Comments and Suggestions for Authors

I am very supportive of the approach outlined in this paper on further dissecting the interaction between viral protein and host protein as a leading opportunity for better opportunities to understand pathogenesis mechanisms and uncover novel targets for therapeutics.  There is much to be done in the area of Host-based Therapeutic approaches, which has been only recently an area of increased scrutiny with the advent of so many more tools in molecular biology.  Good overall review also of PPI for arboviruses and host proteins related to cytoskeleton. 

This paper is a step in that direction as pertains to the large group of Arboviruses, which the author correctly points has a major impact on human health and which appears poised to increase in this impact as a number of climatic and societal influences converge. 

Some thoughts on possible improvements:

line 26 - I question the phrase "billions of cases" and is this a direct quote from one of the references.  If it is, I would list the actual reference, because this seems like an excessive claim.  I would find another way otherwise to signify the magnitude if in fact the term billions cannot be further substantiated.

line 126 and line 130 - Beginning each paragraph with the term "notably".  I would drop the term from line 130.

I will say that there are many instances where the use of a leading word at the start of sentences, sometimes with several in a row, becomes a distraction and often does not substantially add to the sentence itself.  See how many of the catch words such as "Furthermore", "Remarkably", "Notably",  "Interestingly", etc. you can eliminate which will enhance readability.

line 165 - is the term "univocal" meant to convey that these interactions are unambiguous, as in presenting solid evidence of interspecies differences, which is what I believe you are inferring from rat vs. human responses to TBEV virus infection in astrocytes of rats vs. human glioblastoma cells. 

line 171 - I am confused by the sentence "Infected neurons change typical of apoptosis....."  Is the meaning here that virus interaction described in the preceeding sentence supporting the direct effect of viral interaction or the action of an uncontrolled immune response, or both? 

line 175 - I think you have a typographic error for the term neuropathogenesis.

line 185 - "suggested word change for "host cytoskeleton may help reveal elements of their pathogenic properties"

 line 188-190 - suggest also adding the function that MT's are essential for movement of intracellular materials.

line 193 - suggested change to the following:  "However it has yet to be solidly demonstrated that the role played by the cytoskeleton during viral infection is clear...."

Figure 1 - Very useful graphic.  I suggest one enhancement, if it is not too onerous and that is to place large bubbles around each of the three viral impact mechanisms you nicely lay out in the graphic.  For example, it took me a bit of effort to connect up the GTPase activator and molecular complex on the right hand side of the graphic with the fact that it was related to mechanism A.  

line 280 - Suggested change to sentence to highlight what I believe is the main point of the sentence:  ....studies identifying virus-host protein interactions may provide insights into the use of host proteins to assist the virus in replication and immune evasion...."

line 284 - suggested change to sentence to: "...infection induces an early shutoff of host cytoskeletal protein expression..." if this is the intent of the sentence, otherwise it just says that virus shuts off host protein expression.

One thing that I wonder about in all of the cited literature regarding remodeling or repurposing the cytoskeletal architecture is whether there are any morphological studies, with something like confocal microscopy showing physical differences in infected vs uninfected cells.  That would be very convincing, if such exists in the literature.

line 326 - I would say that Arboviruses have an increased likelihood of faster adaptation to environmental and other changes....

line 327 - suggested wording "This takes place under the current state of a lack of specific antiviral....."

line 337 to 341 - there is a missed opportunity here to make a more specific statement about the need for increased discovery of host-based therapeutic approaches to antiviral drugs.  It could be supported by literature citations, such as Pohl et al, 2022. ACS Infectious Dis doi: 10.1021/acsinfecdis.2c00008 or Shaban et al 2022 Trends in Pharmacol Science doi: 10.1016/j.tips.2022.04.004.  These are focused on coronavirus, but the implication is clear. 

Author Response

Reviewer 1

Comments and Suggestions for Authors

I am very supportive of the approach outlined in this paper on further dissecting the interaction between viral protein and host protein as a leading opportunity for better opportunities to understand pathogenesis mechanisms and uncover novel targets for therapeutics.  There is much to be done in the area of Host-based Therapeutic approaches, which has been only recently an area of increased scrutiny with the advent of so many more tools in molecular biology.  Good overall review also of PPI for arboviruses and host proteins related to cytoskeleton. 

This paper is a step in that direction as pertains to the large group of Arboviruses, which the author correctly points has a major impact on human health and which appears poised to increase in this impact as a number of climatic and societal influences converge. 

ANSWER: I thank the Reviewer for the timely review and the valuable suggestions provided, which have hugely improved the content and clarity of the article.

Some thoughts on possible improvements:

line 26 - I question the phrase "billions of cases" and is this a direct quote from one of the references.  If it is, I would list the actual reference, because this seems like an excessive claim.  I would find another way otherwise to signify the magnitude if in fact the term billions cannot be further substantiated.

ANSWER: The relevant bibliographical references have been included in the revised version of the manuscript (see line 27).

line 126 and line 130 - Beginning each paragraph with the term "notably".  I would drop the term from line 130.

ANSWER: I agree. The mentioned term has been eliminated.

I will say that there are many instances where the use of a leading word at the start of sentences, sometimes with several in a row, becomes a distraction and often does not substantially add to the sentence itself.  See how many of the catch words such as "Furthermore", "Remarkably", "Notably",  "Interestingly", etc. you can eliminate which will enhance readability.

ANSWER: As suggested, the above-mentioned redundant terms have been eliminated.

line 165 - is the term "univocal" meant to convey that these interactions are unambiguous, as in presenting solid evidence of interspecies differences, which is what I believe you are inferring from rat vs. human responses to TBEV virus infection in astrocytes of rats vs. human glioblastoma cells. 

ANSWER: Thank you for this observation. I agree that it was unclear that I intended to emphasize the peculiar nature of the virus-host cell interaction, which varies in expression modality depending on the host cell in question. I have amended the text accordingly in the revised version (see lines 102-104).

line 171 - I am confused by the sentence "Infected neurons change typical of apoptosis....."  Is the meaning here that virus interaction described in the preceeding sentence supporting the direct effect of viral interaction or the action of an uncontrolled immune response, or both? 

ANSWER: In the revised text, I have better explained that the observed effects on neuronal cells are a consequence of the direct role of the virus and the activation of the immune system (see lines 108-110).

line 175 - I think you have a typographic error for the term neuropathogenesis.

ANSWER: I apologize. The error has been corrected.

line 185 - "suggested word change for "host cytoskeleton may help reveal elements of their pathogenic properties"

ANSWER: The sentence has been clarified (see lines 125-127).

 line 188-190 - suggest also adding the function that MT's are essential for movement of intracellular materials.

ANSWER: The suggested text integration has been made (see line 131).

line 193 - suggested change to the following:  "However it has yet to be solidly demonstrated that the role played by the cytoskeleton during viral infection is clear...."

ANSWER: The suggested correction has been made (see lines 136-139).

Figure 1 - Very useful graphic.  I suggest one enhancement, if it is not too onerous and that is to place large bubbles around each of the three viral impact mechanisms you nicely lay out in the graphic.  For example, it took me a bit of effort to connect up the GTPase activator and molecular complex on the right hand side of the graphic with the fact that it was related to mechanism A.  

ANSWER: Thank you for this suggestion. The three main viral impact mechanisms have been delimited by a dashed line of a different colour.

line 280 - Suggested change to sentence to highlight what I believe is the main point of the sentence:  ....studies identifying virus-host protein interactions may provide insights into the use of host proteins to assist the virus in replication and immune evasion...."

line 284 - suggested change to sentence to: "...infection induces an early shutoff of host cytoskeletal protein expression..." if this is the intent of the sentence, otherwise it just says that virus shuts off host protein expression.

ANSWER: Thank you for the above suggestions. The changes have been made (see lines 232-234 and 236-238).

One thing that I wonder about in all of the cited literature regarding remodeling or repurposing the cytoskeletal architecture is whether there are any morphological studies, with something like confocal microscopy showing physical differences in infected vs uninfected cells.  That would be very convincing, if such exists in the literature.

ANSWER: Some studies carried out by using the confocal microscopy approach have been mentioned in the revised text (see lines 215-220).

line 326 - I would say that Arboviruses have an increased likelihood of faster adaptation to environmental and other changes....

ANSWER: The suggested text integration has been made (see lines 283-285).

line 327 - suggested wording "This takes place under the current state of a lack of specific antiviral....."

ANSWER: The suggested text integration has been made (see lines 285-286).

line 337 to 341 - there is a missed opportunity here to make a more specific statement about the need for increased discovery of host-based therapeutic approaches to antiviral drugs.  It could be supported by literature citations, such as Pohl et al, 2022. ACS Infectious Dis doi: 10.1021/acsinfecdis.2c00008 or Shaban et al 2022 Trends in Pharmacol Science doi: 10.1016/j.tips.2022.04.004.  These are focused on coronavirus, but the implication is clear. 

ANSWER: I am very grateful for this suggestion, that has been inserted in the text (see lines 297-306).

Reviewer 2 Report

Comments and Suggestions for Authors

In this paper, the author wants to highlight the mechanisms known to reorganize and manipulate the cytoskeleton during infection and how this could contribute to pathogenesis. The references are well presented, however there are some concerns regarding the manuscript and some suggestions to increase overall manuscript quality.

The author states that "an accurate, in-depth analysis of the host cell factors co-opted during the infectious process and the mechanisms by which they cause neural dysfunction is lacking.". Besides presenting some evidence present in the literature of cytoskeleton involvment, the manuscript lacks a strong discussion of findings in the present literature. How exactly the cytoskeleton is being related to the pathogenesis? Together, these evidences present any distinct pattern? Different types of filament are modulated through different types of viruses? What are the modulation differences among neurotropic and non-neurotropic but neuroinvasive viruses? Some descriptions mention the viral proteins involved while the other not. For viral infections where the relevant protein is not cited it is because it is not described so far? It was purposely supressed? It is problematic that the paper starts by the affirmation that in-depth analysis is needed to compile all available evidence and ends without a proper discussion of all the evidences mentioned. This impairs the possibility of a relevant contribution of the manuscript to the field.

The author quickly cites a very similar paper that she is first author published in this year, in february, available in the link: https://link.springer.com/article/10.1007/s40588-025-00241-4

Despite the basic differences (CS reorganization and modulation by all viruses vs. by arboviruses), there are many quotes and parts that are very similar. Even the figure itself has many common graphical parts but without deeper elaboration in the present paper.

The idea of discussing only what is known about arboviruses is important, however it is not achieved with the expected relevant quality to improve researchs in the field, ultimately leading to a expanded simplified version of the first article. Additionally, there are some considerations to increase overall quality in future submissions:

Line 12. Avoid sentences like “We still know little about the host interactions with arboviruses.”

Line 42. “an accurate, in-depth analysis of the host cell factors co-opted during the infectious process and the mechanisms by which they cause neural dysfunction is lacking”. There are many different analysis of mechanism. Author can state that a review is lacking.

Line 80. In terms of acute disease management use the term “treatment” instead of “cure”

Line 83. Culex means the genus or multiple species? Please refer it as Culex genus or Culex spp.

Line 84. JEV lacks reference supporting acute encephalitis

The author wrote almost three pages introducing the viruses that are going to be presented in the paper and how they are related to neuroinvasion. This is exhaustive and hard to track over the reading. I suggest to replace paragraphs of viruses description by a large table. Easier and practical to presente the 13 different viruses in the study.

Line 182. Avoid sentences like “Little is known about the molecular mechanisms underlying the cell biology of arbovirus infection.” There is plenty of knowledge about cell biology during viral infections. This statement is not true and is not verifiable.

Line 269. Avoid using terms like “The data above.” The author is not presenting any data, just summaries from other authors reports

Line 284 and 292. “Up-regulation” and “upregulation”. Occurs other times throughout the text.

Line 291. List the genes involved in the processes mentioned. It is important for further reading and comparisons with other articles.

Line 309. What does the expression "These issues may offer..." means? It is confusing and misleading as it sounds as if the evidence that points to the result is problematic.

Table 1. The author creates a table to describe proteins and pathways associated with CS modulations using only 3 different references. More references are needed to be provided, or the whole table can be collapsed to a single paragraph.

Line 343. Again, avoid expressions like “The data shown here attest…”

Comments on the Quality of English Language

The aren't any noticeable grammar errors, but the English quality of the text could be improved. There are many sentences that do not reflect the overall objective of this manuscript, suggesting that there is a lack of evidence in the field, which is not true. Some words should be avoided in scientific communications and must also be reconsidered.

Author Response

Reviewer 2

In this paper, the author wants to highlight the mechanisms known to reorganize and manipulate the cytoskeleton during infection and how this could contribute to pathogenesis. The references are well presented, however there are some concerns regarding the manuscript and some suggestions to increase overall manuscript quality.

ANSWER: I thank the Reviewer for the timely review and the valuable suggestions provided, which have hugely improved the content and clarity of the article.

The author states that "an accurate, in-depth analysis of the host cell factors co-opted during the infectious process and the mechanisms by which they cause neural dysfunction is lacking.". Besides presenting some evidence present in the literature of cytoskeleton involvment, the manuscript lacks a strong discussion of findings in the present literature. How exactly the cytoskeleton is being related to the pathogenesis? Together, these evidences present any distinct pattern? Different types of filament are modulated through different types of viruses? What are the modulation differences among neurotropic and non-neurotropic but neuroinvasive viruses? Some descriptions mention the viral proteins involved while the other not. For viral infections where the relevant protein is not cited it is because it is not described so far? It was purposely supressed? It is problematic that the paper starts by the affirmation that in-depth analysis is needed to compile all available evidence and ends without a proper discussion of all the evidences mentioned. This impairs the possibility of a relevant contribution of the manuscript to the field.

ANSWER: I thank the Reviewer for the critical evaluation of the text content. I have reported the above-mentioned concepts in the Discussion section (see lines 307-345).

The author quickly cites a very similar paper that she is first author published in this year, in february, available in the link: https://link.springer.com/article/10.1007/s40588-025-00241-4

Despite the basic differences (CS reorganization and modulation by all viruses vs. by arboviruses), there are many quotes and parts that are very similar. Even the figure itself has many common graphical parts but without deeper elaboration in the present paper.

ANSWER: The concerns raised by the Reviewer have been punctually explained in a dedicated rebuttal sent to the Editor upon the request for clarification.

The idea of discussing only what is known about arboviruses is important, however it is not achieved with the expected relevant quality to improve researchs in the field, ultimately leading to a expanded simplified version of the first article. Additionally, there are some considerations to increase overall quality in future submissions:

Line 12. Avoid sentences like “We still know little about the host interactions with arboviruses.”

ANSWER: The referenced sentence has been revised.

Line 42. “an accurate, in-depth analysis of the host cell factors co-opted during the infectious process and the mechanisms by which they cause neural dysfunction is lacking”. There are many different analysis of mechanism. Author can state that a review is lacking.

ANSWER: The text has been revised accordingly (see lines 41-43).

Line 80. In terms of acute disease management use the term “treatment” instead of “cure”

ANSWER: The suggested correction has been made.

Line 83. Culex means the genus or multiple species? Please refer it as Culex genus or Culex spp.

ANSWER: The suggested correction has been made.

Line 84. JEV lacks reference supporting acute encephalitis

ANSWER: References have been inserted in the revised version of the manuscript (see the new Table 1).

The author wrote almost three pages introducing the viruses that are going to be presented in the paper and how they are related to neuroinvasion. This is exhaustive and hard to track over the reading. I suggest to replace paragraphs of viruses description by a large table. Easier and practical to presente the 13 different viruses in the study.

ANSWER: I thank the Reviewer for the suggestion. The text related to the neurotropic arbovirus description has been replaced by the new Table 1.

Line 182. Avoid sentences like “Little is known about the molecular mechanisms underlying the cell biology of arbovirus infection.” There is plenty of knowledge about cell biology during viral infections. This statement is not true and is not verifiable.

ANSWER: The mentioned sentence has been modified (see lines 123-124) .

Line 269. Avoid using terms like “The data above.” The author is not presenting any data, just summaries from other authors reports

ANSWER: The sentence has been modified (see line 221).

Line 284 and 292. “Up-regulation” and “upregulation”. Occurs other times throughout the text.

ANSWER: The errors have been amended.

Line 291. List the genes involved in the processes mentioned. It is important for further reading and comparisons with other articles.

ANSWER: The gene list has been inserted in the revised manuscript (see lines 242-245).

Line 309. What does the expression "These issues may offer..." means? It is confusing and misleading as it sounds as if the evidence that points to the result is problematic.

ANSWER: The referenced sentence has been revised.

Table 1. The author creates a table to describe proteins and pathways associated with CS modulations using only 3 different references. More references are needed to be provided, or the whole table can be collapsed to a single paragraph.

ANSWER: One of the limitations of this review is related to the fact that, to date, there are very few data in the literature regarding experimental infections conducted in vitro or in vivo with neurotropic strains of arboviruses. Therefore, the table reports the available information, and the aforementioned limitation has been highlighted in the Discussion section of the article.

Line 343. Again, avoid expressions like “The data shown here attest…”

ANSWER: The sentence has been modified.

Comments on the Quality of English Language

The aren't any noticeable grammar errors, but the English quality of the text could be improved. There are many sentences that do not reflect the overall objective of this manuscript, suggesting that there is a lack of evidence in the field, which is not true. Some words should be avoided in scientific communications and must also be reconsidered.

ANSWER: The article was reviewed by a native English-speaking Professor.